# Endocrine Disrupting Compounds (Nonylphenol and Bisphenol A)–Sources, Harmfulness and Laccase-Assisted Degradation in the Aquatic Environment

**DOI:** 10.3390/microorganisms10112236

**Published:** 2022-11-11

**Authors:** Agnieszka Gałązka, Urszula Jankiewicz

**Affiliations:** Department of Biochemistry and Microbiology, Institute of Biology, Warsaw University of Life Sciences-SGGW, Nowoursynowska 159, 02-787 Warsaw, Poland

**Keywords:** bioremediation, endocrine disrupting compounds, laccase, pollution, xenoestrogens

## Abstract

Environmental pollution with organic substances has become one of the world’s major problems. Although pollutants occur in the environment at concentrations ranging from nanograms to micrograms per liter, they can have a detrimental effect on species inhabiting aquatic environments. Endocrine disrupting compounds (EDCs) are a particularly dangerous group because they have estrogenic activity. Among EDCs, the alkylphenols commonly used in households deserve attention, from where they go to sewage treatment plants, and then to water reservoirs. New methods of wastewater treatment and removal of high concentrations of xenoestrogens from the aquatic environment are still being searched for. One promising approach is bioremediation, which uses living organisms such as fungi, bacteria, and plants to produce enzymes capable of breaking down organic pollutants. These enzymes include laccase, produced by white rot fungi. The ability of laccase to directly oxidize phenols and other aromatic compounds has become the focus of attention of researchers from around the world. Recent studies show the enormous potential of laccase application in processes such as detoxification and biodegradation of pollutants in natural and industrial wastes.

## 1. Introduction

Pollution of the aquatic environment with substances such as pesticides, heavy metals, polycyclic aromatic hydrocarbons, microplastics, and pharmaceuticals enter waters as a result of anthropogenic activities and endanger the health of plants, animals, and humans due to their acute toxicity and potential accumulation. Endocrine disrupting compounds (EDCs) are substances of considerable risk to human health. The list of EDCs is rapidly growing. According to the TEDX (The Endocrine Disruption Exchange) database, the number of suspected EDCs was 881 in 2011, increasing to 1419 in 2017 [1].

EDCs include a wide range of chemicals that are present in people’s daily lives as ingredients in everyday items, cleaning products, and medications. In 2015, the total production of chemicals within the 28 member states of European Union (EU) was 323 million tons, 205 million tons of which were considered hazardous to health [2].

Ismail et al. in 2017 differentiated the EDCs natural and synthetic compounds, such as hormones, alkylphenols (AP), polyhalogenated compounds, bisphenol A (BPA), phthalates, pharmaceuticals, and pesticides. These EDCs are released into the environment from various sources [3]. An especially dangerous group of waste are the alkylphenols (AP), because of their interference with the proper functioning of the endocrine system in humans and animals. A typical representative of this group of compounds are nonylphenols (NPs) and bisphenols (BPs)-xenoestrogens used in the production of products commonly used in households [4].

Determining the key features of the EDCs by Merrill et al. in 2020 will facilitate the evaluation of chemicals in terms of their effects on the endocrine system. At this point, it should be noted that since the hormones generally act throughout the entire systems, it takes only one specific action by any given compound to disrupt the entire system [5]. EDCs function as agonists or antagonists, i.e., they increase or inhibit the estrogenic activity of cells. The type of action may depend on the concentration of a given substance in the environment [6].

The literature on the molecular and clinical effects of hormones and EDC is extremely extensive, but even in the case of extensively studied BPA, there are still gaps in identifying the key chemical properties/mechanisms that lead to the manifestation of phenotypic changes in humans and animals [5].

Aquatic organisms, including fish, are particularly exposed to EDCs. Exposure to EDCs may result in abnormal vitellogenin induction, altered sex determination, decreased growth rate, reproductive delay, and altered behaviors [7,8].

The rapidly growing human population and aging society (up from 629,800 in 2016 to almost 1.6 million in 2051) has led to an increased in both the use and variety of medications consumed, thereby increasing the variety of pharmaceuticals entering surface waters [9]. The presence of alkylphenols like nonylphenol (NP) and BPA in aquatic environment closely correlates with anthropogenic activity.

Nonylphenol (NP) is a xenobiotic compound consisting of a phenolic ring and a nine-carbon chain in the para position. Under ambient conditions, it is a viscous liquid of light pale color, immiscible with water. In industry, NP is produced by alkylation of phenol with nonene under acid catalysis. The final technical mixture consists of more than 22 isomers of 4-substituted alkylphenols [10]. The first evidence that alkylphenols can be estrogenic was published in 1938 by Dodds and Lawson [11]. NP is used as surfactants. It is well soluble in methanol and ethanol, and insoluble in water. It is characterized by high resistance to degradation processes, while showing a tendency to bioaccumulation. In 2008, 4-NP was included in the list of priority substances in the field of water policy included in the Directive of the European Parliament and of the Council (2008/105 /EC) [12], and has been identified as a priority hazardous substance.

Bisphenol A (BPA) was synthesized in 1891 and has been used since the 1960s for the interior coating of plastics and cans, making it one of the most produced compounds [13]. BPA in polycarbonate plastics and epoxy resins is one of the main sources of BPA in water [14]. Based on the physicochemical characteristics of BPA, it is estimated that its particles in the environment are bound to water and suspended solids (53%), soil (25%), or sediment (23%) [14,15,16].

The presence of BPA in surface and groundwater, even at ppb concentration, is considered harmful to the environment. There are BPA residues in natural ecosystems that damage the environment and metabolism [13,17,18,19,20]. BPA, like NP, tends to accumulate in aquatic animals; hence its bioaccumulation along the surface water food chain [21,22,23,24]. Barboza et al. found that the levels of BPA in the liver and muscles of various North-East Atlantic fish were 3,9 ± 14.0 ng/g and 16.2 ± 48.3 ng/g, respectively [25]. Human consumption of BPA-containing marine fish can cause BPA to be present in the human body.

Based on EC50 and LC50 values, which range from 1 to 10 mg/L, BPA has been classified as “moderately toxic” and “toxic” to aquatic fauna by the European Commission and the United States Environmental Protection Agency (US EPA) [14].

Studies conducted in different countries have proved the presence of these substances in influent, but also in wastewater treatment plants and effluent. The presence of EDCs like NP and BPA in water reservoirs, despite low concentrations of a few ng/L, is an elevated risk of undesirable effects for human physiology, but mainly for aquatic organisms.

EDCs are not removed during wastewater treatment and may persist in surface waters below discharges from wastewater treatment plants [26,27,28,29,30,31]. Biodegradation is the major removal mechanism that affects the fate and transport of estrogenic compounds in environments [32]. Laccases have gained significant attention due to their emerging applications including bioremediation, biomass degradation, and biofuel cells. In bacteria, the researchers found low-potential laccases, with a redox potential from 0.36 V to 0.46 V, and in fungi (mainly basidiomycetes) high-potential laccases with a redox potential >0.71 V was found. High-potential laccases possess a broader substrate spectrum than low-potential, including EDCs. In aquatic environments, laccase use water as co-substrate, because of which they are often called “green” biocatalysts [33].

In this review, we focused on presenting the structure, physicochemical properties, and the presence of NP and BPA as representatives of EDCs in various aquatic environment matrices, in water and in sediments, and the correlation between the above-mentioned features and the difficulties in removing these compounds from the ecosystem. Studies have found that biodegradation is one of the promising methods of removing EDCs from the environment. In this article, we present the possibility of using microorganisms producing laccase and the enzyme itself in the biodegradation of NP and BPA, considering immobilization as one of the ways to improve enzymatic properties.

## 2. Endocrine Disrupting Compounds (EDCs)–Structure, Characteristics and Risk

The endocrine disrupting compounds (EDCs) are exogenous chemicals that disrupt the function of hormones. Hormones secreted by glands into the blood interact with specific receptors in the cells of target organs. These interactions lead to the regulation of human physiological processes, including the maintenance of holistic homeostasis, growth, development, reproduction, energy balance, metabolism, and weight regulation. Exogenous chemicals can disrupt this complex communication system, thereby increasing the risk of adverse health effects, including cancer (e.g., breast, ovary, uterus), reproductive impairment, neurodegenerative disease, metabolic diseases, e.g., diabetes, cardiovascular disease, and obesity [5,6].

Endocrine disrupting chemicals are a highly heterogeneous group of molecules with strong estrogenic activity [34]. EDCs can disrupt the normal function of the endocrine system of vertebrate and invertebrate organisms by: (i) mimicking the action of endogenous hormones; (ii) antagonizing the action of endogenous hormones; (iii) disrupting the synthesis and metabolism of endogenous hormones or (iv) disrupting the synthesis of certain hormone receptors [3,6,35]. Through the above-mentioned mechanisms, EDCs disrupt the production, release, transport, and metabolism, binding, action, and elimination of natural hormones in the body [6].

Various classes of compounds disrupt the endocrine systems. A common feature of most EDCs is the possession of at least one aromatic group in their structure [36]. Kiyama et al. in 2015 proposed the division of EDCs due to the structure of these chemical compounds and their application. By structure, we can divide EDCs into phenols and non-phenols. Phenolic EDCs include simple phenols, phenolic acids/phenolic aldehydes, acetophenones, tyrosine derivatives, phenylacetic acids, hydroxycinnamic acids, phenylpropenes, coumarins/isocoumarins/chromones, naphthoquinones, bisphenols, benzophenones/chalcones, stilbenes/stilbenoids, flavones/flavonoids, lignans/neolignans, diarylheptanoids, polycyclic aromatic hydrocarbons (PAHs) and hydroxylated PAHs. Non-phenols include anilines, carboranes, indoles, metalloestrogens, perfluorinated compounds, phthalates and terpenes/terpenoids (monoterpenes, sesquiterpenes, diterpenes, triterpenes, tetraterpenes, sterols, saponins and meroterpenes) [6].

Due to its application, Kiyama et al. in 2015 listed food additives, dietary supplements, pesticides (antimicrobials/essential oils, biocides, disinfectants/disinfectants, fungicides, herbicides, insecticides, soothing agents, repellants, rodenticides and pheromones), pharmacological estrogens (e.g., zearalenone, non-steroidal pharmaceutical ER antagonists, SERMs and steroids), plasticizers (phthalate/trimellitate/salicylate), organic esters and phosphates) and waste (heavy metals or metalloestrogens, persistent organic pollutants or POPs, pesticide residues, pharmaceutical contaminants and PAHs) [6].

La Merrill et al. in 2020 proposed ten key features of EDCs. The key features of EDCs include the main mechanisms by which endocrine systems can be disrupted. Based on the knowledge of the mechanisms by which hormones and EDCs exert their specific effects, the following characteristics of EDCs are listed: (i) interaction with or activation of hormone receptors (ii) antagonization of hormone receptors (iii) alteration of hormone receptor expression (iv) alteration in signal transduction (including changes in protein or RNA expression, post-translational and/or ion flux modifications) in hormone responsive cells (v) induction of epigenetic modifications in hormone producing or hormone responsive cells (vi) alteration of hormone synthesis (vii) alteration transport of hormones across cell membranes (viii) alteration of hormone distribution or circulating hormone levels (ix) alteration of metabolism or clearance of hormones (x) alteration of the fate of hormone producing or hormone responsive cells. The activities of EDCs depicted include enhancement and weakening of effects. Various well-known EDCs show different interference characteristics with endocrine systems [5].

The mechanism of estrogen signaling is complex, and involves intracellular and extracellular signaling networks. The intracellular network includes genomic and non-genomic pathways. In the genomic pathway, transcription of target genes is altered by the binding of chemicals (or ligands) to the nuclear ER, ERα and Erβ. In the non-genomic pathway, signal transduction is started by ligand binding to membrane/cytoplasmic ER receptors and/or other receptors, e.g., GPER, ER-X. Autocrine and/or paracrine signaling pathways include other hormones, growth factors, and cytokines. Intracellular networks cooperate with diverse types of autocrine and/or paracrine signaling, and thus cells in different tissues or locations are also involved in estrogen signaling pathways e.g., estrogens, by influencing the synthesis and secretion of growth hormone (GH) or insulin growth factor, can influence somatic growth [37].

NP and BPA, chemicals belongings to alkylphenols, were selected for further analysis showing a strong relationship between exposure and endocrine effects in humans, wild animals, or aquatic organisms. Both are quite common in the aquatic environment as organic pollutants of anthropogenic origin and are biodegradable by both fungal and bacterial laccase.

### 2.1. Nonylphenol

Nonylphenol (NP) belongs to the group of alkylphenol compounds (APEs) of the family of non-ionic surfactants, and it was produced for the first time in 1940 [38]. It is a xenoestrogen formed during degradation of ethoxylates nonylphenol (NPEO) [39,40].

4-NP is the most common commercial form of NP and is used in experiments and analyzes. NP is persistent, lipophilic, and tends to bioaccumulate more than NPEO [41]. Nonylphenol is a hydrophobic compound with octanol-water partition coefficient (log K_OW_) of 4.48 and low water solubility. The result is low NP mobility and a significant reduction in distribution in the aquatic environment [41]. NP is a semi-volatile organic compound capable of binding water. Once nonylphenol enters the atmosphere, it can be reintroduced into the surface water ecosystem with precipitation [42]. At room temperature, NP is a light-yellow liquid with an approximate molecular weight of 215 to 220 g/mol and a specific weight of 0.953 g/mL at 20 °C. It has a dissociation constant (pKa) of 10.7 ± 1.0 and a log K_OW_ between 3.8 and 4.8, and shows both pH and temperature dependent solubilities, showing values of 6350 μg/L at pH 5 and 25 °C [41].

NP is a compound that has isomers consisting of nine-carbon alkyl chains attached to a phenolic ring. A hydroxyl group is present on the phenolic ring of NP. The isomers differ in the carbon atom in the phenolic ring to which the alkyl chain is attached. Isomers include compounds with phenolic substitution at the meta-, ortho-and para positions (called 2-, 3-, and 4-alkylphenols, respectively). 4-nonylphenols (4-NP) and 4-octylphenols (4-OP) account for over 80% of total APE production. The alkyl group is branched or linear. It should be noted that shorter chain NP isomers (i.e., 4-n-NP) are more resistant to degradation than branched isomers and persist in sediments. The half-life of NPs in sediments has been found to be over sixty years [43,44].

Due to its structural similarity, NP mimics the natural hormone 17β-estradiol and competes with it for binding sites in the receptors, although with a lower affinity than the natural hormone [45,46]. The estrogenic effects of NP vary depending on the isomers, with a high estrogenic effect observed with 4-(1′,1′-dimethyl-2′-ethylpentyl)-phenol (NP7) [47]. Not all nonylphenol isomers are capable of inducing the estrogenic activity.

NP also has an anti-androgenic effect, i.e., it may interfere with the proper functioning of androgens necessary for the proper development of men and their reproductive system due to the multi-stage activation of the androgen receptor [48]. As a result of the above mechanism, NP induces disorders in men, including lowering circulating testosterone levels in the blood, decreased activity of antioxidant enzymes in sperm, and disturbed structure of the testes, as well as increased apoptosis of Sertoli cells [49,50,51]. Moreover, it has been shown that high exposure of women to NP in the second trimester of pregnancy led to reduced birth weight and premature deliveries [52,53,54].

NP may cause feminization of aquatic organisms, reduce male fertility, and survival of young animals at a concentration of 8.2 mg/L [46,55]. It has significant acute toxicity to phytoplankton, zooplankton, amphibians, invertebrates, and fish [56]. Male fish have intersex traits, low testosterone values, produce vitellogenin (a female-specific protein), and show gonadal changes that reduce fertility [57,58].

Due to the harmful effects of the degradation products of ethoxylates nonylphenol, the use and production of such compounds has been banned in EU countries. NP and its ethoxylates have been identified as priority hazardous substances (PHS) in the Water Framework Directive (Directive 2000/60/EC, 2000) and most of their uses are now regulated (Directive 2003/53/EC, 2003) [46]. Maximum allowable concentrations of NP in the EU in freshwater is 0.3 μg/L, in sediments of freshwater reservoirs 0.18 mg/kg dry weight, no limits for sea water [44].

The United States Environmental Protection Agency (EPA) recommends a nonylphenol concentration in freshwater below 6.6 μg/L and in salt water below 1.7 μg/L TDI for NP is 5 µg/kg body weight/day [46].

Due to the lack of NP mineralization in anaerobic conditions, these compounds are subject to bioaccumulation in the environment and may reach higher concentrations in next trophic levels. This is more likely for benthic organisms in close contact with contaminated sediment [59,60]. NP bioaccumulation has been noted in algae, fish, and aquatic birds living in the environment surrounding the contaminated river [61,62,63]. The presence of nonylphenol in fish is usually associated with wastewater discharge from sewage treatment works (STW), leading to concentrations of up to 110 μg/kg in fish [64,65].

Aquatic organisms are susceptible to the release of new chemicals into the environment because the recipients of wastewater from wastewater treatment plants are rivers, estuaries, and oceans. Even though nonylphenol is excreted from the tissues of aquatic species, these organisms may be continuously exposed to chemicals throughout their lives [34,39,40].

### 2.2. BPA

Bisphenol A (BPA) is one of the most common pollutants found in water bodies. In chemical terms, BPA is an organic synthetic compound that belongs, like NP, to the group of alkylphenols. 2,2- (4,4′-dihydroxy diphenyl) propane is the full name of BPA (molecular formula C_15_H_16_O_2_; M_w_ = 228.29 g/mol). BPA is a structural analogue of bisphenol that is 4,4′-methanediyldiphenol in which the methylene hydrogens are replaced by two methyl groups. This compound is obtained by combining 2 moles of phenol with 1 mole of acetone [66].

BPA has a low vapor pressure, high melting point, and moderate solubility [14]. The water solubility is 300 mg/L at 25 °C. The physicochemical properties of BPA indicate that it has a low rate of evaporation from soil and water.

BPA has low or moderate hydrophobicity, log K_OW_ 3.32; organic carbon partition coefficient (log K_OC_) ranging from 2.50 to 4.5 showing a moderate mobility and bioaccumulation potential of this compound [14,67]. However, mobility can be affected by soil chemistry and texture. Some studies confirm that increased BPA sorption occurs in the presence of iron, cadmium, and lead [68,69] and in sandy, acidic soils there is a rapid and complete desorption of BPA [70].

BPA is a solid and is commercially sold in the form of crystals, pills, or flakes. When BPA melts at elevated temperature during production, particles released into the environment are typically dissolved in water [16].

BPA is associated with an approximately 10,000–100,000 times weaker for ERβ than that of estradiol, so it was considered a very weak environmental estrogen [71,72]. Although BPA has been shown to have weak estrogenic activity, it can disrupt the proper functioning of the endocrine system even at extremely low concentrations [73]. Prenatal exposure to BPA may increase the tendency to develop breast cancer in adulthood [74].

Since BPA in the human body is rapidly absorbed through the stomach and intestines, the major glucuronic acid metabolite is formed in the liver and then excreted in the urine with a short half-life. BPA exposure is usually determined by analyzing urine samples. Screening studies have shown that the urine of most people in industrialized countries contains measurable levels of BPA and its metabolites [75,76]. The human body contains two types of BPA, unconjugated BPA and conjugated BPA. Unlike unconjugated BPA, conjugated BPA has no estrogenic activity and is harmless to humans. Absorbed BPA is metabolized in the liver to conjugated BPA. When BPA enters the body through skin, bypassing the digestive system, more unconjugated BPA circulates in the blood [77,78,79].

Studies have showed the effect of BPA on the development of invertebrates. Both the midge larvae (*Chironomus riparius*) and the sea copepod (*Tigriopus japonicus*) showed growth inhibition at exceptionally low BPA concentrations (0.08 and 0.1 mg/L, respectively). At concentrations from 1.1 to 12.8 mg/L BPA is systemically toxic to various taxa, including daphnia, mice, freshwater fish (*Pimephales promelas*), and saltwater fish (*Menidia menidia*) [14]. The effects of BPA appear to vary significantly among related taxa, and it appears that invertebrates may be hypersensitive to BPA exposure (in particular freshwater molluscs and insect larvae, and marine copepods in particular).

In aquatic vertebrates, exposure to BPA has effects on fish reproduction due to alteration of sex cell proportions, masculinization or feminization, structural changes in gonads, reduction of sperm quality, and delay or inhibition of ovulation, and induction of VTG (egg yolk protein precursor). The VTG protein is a widely used biomarker of vertebrate exposure to estrogenic compounds [25,80,81].

Assessment of the effects of exposure to BPA on wild mammals is currently based on data from laboratory studies in model organisms which show harmful effects on rodents at high BPA levels. Such effects include faster maturation, increased obesity, complications of pregnancy, malformations of male and female reproductive organs, impact on prostate, and increase in cancer incidence [66,72,82,83]. BPA can cross the placenta during pregnancy and accumulate in the amniotic fluid and fetal plasma [84]. BPA is listed as class 1B toxic for reproduction by the European CLP regulation [85,86].

BPA toxicity is expressed through endocrine disruption, but also through neurotoxicity, cytotoxicity, reproductive toxicity, genotoxicity, and carcinogenicity [79]. Due to the harmful effects on human and animal health, checking the levels of this compound and structural analogues in food and beverages is a priority. Therefore, the U.S. Environmental Protection Agency (EPA) has set the largest Tolerable Daily Intake (TDI) for BPA at 50 μg/kg body weight per day, and the oral reference dose (RfD) for BPA as a total allowable concentration (TAC) in drinking water is 100 μg/L. The EU has also introduced similar regulatory measures [44].

Published papers confirm the operation of BPA as EDCs. The BPA molecule is structurally similar to steroid hormones and has an estrogen-mimicking effect (E2). Studies show that BPA exerts an agonistic effect on both types of estrogen receptors (ERα and ERβ) and on G-protein coupled receptors (GPCR) and the gamma estrogen-related receptor (ERRc) [5,71,72,87]

BPA meets nine out of ten key features of EDCs proposed by La Merrill et al. in 2020 [5]. BPA activates nuclear and membrane ERs, GPER, which has multiple effects on organs, antagonizes the androgen receptor, increases ER mRNA expression in specific areas of the mouse brain, induces proliferation Sertoli TM4 cells by inducing ERK phosphorylation, affects promoter-specific methylation in brain, prostate and breast cancer cells, lowers the level of cytochrome P450 aromatase and the expression of other steroidogenic regulatory proteins, reduces insulin secretion from pancreatic β-cell vesicles, increases the level of SHBG in men and lowers the level of circulating androstenedione and free testosterone, and developmental exposure to BPA increases the proliferation index in the mammary gland and pancreas endothelial cells of the uterus [44]. Chronic diseases such as prostate and breast cancer, type 2 diabetes, obesity, and brain development disorders occur due to early exposure to BPA [88,89,90].

As a result of the xenoestrogen activity, BPA can activate immune effects mediated by estrogens, and thus can induce pro-inflammatory pathways. By acting on estrogen receptors, BPA also affects the immune system, e.g., changing the function of dendritic cells and T and B lymphocytes and can induce production of reactive oxygen species by neutrophils [91].

Overall, its action is based on different molecular and epigenetic mechanisms that converge in the endocrine and reproductive systems [72].

## 3. Endocrine Disrupting Compounds (EDCs)–Occurrence and Removal Methods in Water Environment

Environmental pollution has become a major challenge in recent years due to increasing population, urbanization, and industrialization [27]. The priority is to find the sources of EDCs and the routes by which they enter the aquatic environment.

EDC has been detected in a variety of fresh, brackish, and marine ecosystems. Due to their physical and chemical properties, EDCs can bioaccumulate, biomagnify, are persistent, and are very toxic to aquatic organisms, both for plants and animals [3].

These compounds are released into the environment from a variety of sources, primarily municipal and industrial waste, agricultural practices, animal waste, and sewage treatment plants (STP) [92]. Most of the packaging for food, cosmetic products, solvents, preservatives, and pesticides is also made of EDC-containing materials [3,44,93].

Long-term use of EDC, regardless of the concentration level, may accumulate in animals and be partially released into the environment through animal feces. In fish and the largest consumers of the food web, the rate of bioaccumulation is higher because most EDCs are lipophilic and concentrated in the fat of the consuming organisms [11]. Therefore, these substances will penetrate the food chain and ecosystems, potentially adversely affecting human health.

In agriculture, non-metabolized and non-degradable compounds in animal fertilizers still have active metabolites, reducing the quality of surface and groundwater and significantly affecting aquatic life. In water, EDCs undergo biodegradation and chemical and photochemical degradation, dilution, and sorption to sediments, which partially leads to their elimination from aquatic ecosystems [3,94,95,96].

### 3.1. Nonylphenol

The presence of nonylphenol in the environment is clearly correlated with anthropogenic activities such as sewage treatment, storage, and recycling of sewage sludge. NPEOs are used as non-ionic surfactants in industry (cellulose and paper, textiles, agriculture, metals, plastics, petroleum refining), in households in the form of detergents, solubilizers, and personal care products in non-EU countries [38,45]. Nonylphenol is a xenobiotic compound used in the production of antioxidants, additives to lubricating oils and in the production of ethoxylated surfactants nonylphenol, which is its main use (65%) [46].

As surfactants, NPs are used in cleaning agents, so their primary source in the environment are discharges of wastewater from industrial and municipal wastewater treatment plants (WWTPs), as well as land enriched with solid sewage or manure, runoff from pesticides and fertilizers on agricultural fields, and fodder livestock. Due to their cheapness, NPEO surfactants are used in various areas, for example in agricultural pesticides where surfactant is added to control the properties of pesticides [34,97].

Inadequately treated domestic sewage causes high concentrations of NP in the aquatic environment. The levels of NP depended on the size of NP discharges into the river, temperature, flow velocity, biodegradation, etc. About 60% of NP and its derivatives produced in the world ends up in water supply [98,99,100]. In addition, the presence of NP is observed in polyvinyl chloride (PVC), which can contaminate water passing through PVC plumbing [101].

Technical NP is the major form of NP that is released into the environment. Technical nonylphenol consists of a mixture of more than one hundred isomers that have an alkyl moiety attached at various positions on the phenolic ring, with para-substituted NP (4-NP). However, in the environment, the proportions of isomers may be different [10,34,102].

Due to its high hydrophobicity, resistance to biodegradation and low solubility, NP tends to accumulate in various environmental matrices [4]. NP can evaporate into the atmosphere from wastewater discharges, wastewater treatment plant (liquids and sludge) or heavily contaminated surface waters. NP binds to the aerosols generated by the wastewater treatment plant, leading to a reduction in air quality in the vicinity of STW. From the atmosphere, NP may re-enter aquatic and terrestrial ecosystems with rain and snowfall [46].

The concentration of nonylphenol in the surface layers of natural waters may decrease due to photolysis induced by sunlight [94,103]. Biodegradation of NPs is difficult due to physicochemical properties such as low solubility and high hydrophobicity. NP accumulates in environmental compartments that are characterized by a high content of organic substances, usually sewage sludge and river sediments. NP occurs in river waters in concentrations up to 4.1 μg/L and 1 mg/kg in sediments [98,99,104]. The concentration of NP in the water and sediments are shown in Table 1.

**Table 1 microorganisms-10-02236-t001:** The Bisphenol A (BPA) and nonylphenol (NP) concentrations in aquatic environments.

Environment		BPA	NP	Location	References
**Surface water**		nd–376.6	117–865	China, Pearl River Delta	[105,106]
		15.80–110.38	nd–104.02	China, Honghu Lake	[107]
		16.3–30.1	-	Poland, Vistula river	[108]
		0.71–47.40	-	Philippines, Laguna Lake	[109]
**Groundwater**		nd–35.54	5,6	China	[110,111]
		1–629	-	Poland	[108]
**Wastewater treatment plants**	IW	nd–5927.5	nd–9560	Poland	[108]
	EW	nd–190	nd–9560		
	IW	234.6–1527.1	519.6–4183.4	China	[112]
	EW	3.1–623.6	13.4–471.6		
**Seawater**		10.6–52.3	22–201	Greece, Thermaikos Gulf, Northern Aegean Sea	[113]
		240	730	Japan, Tokyo Bay coastal area	[114]
**Sediment**	W	7.02–13.95	-	China, Erhai Lake	[115]
	S	11.09–63.46	-		
	W	-	75.4	Germany, Luppe River	[116]
	S	-	22,342.6		
	W	nd–37,000	-	Indonesia, Mahakam River	[117]
	S	nd–952.6	-		
	W	<1.3–5.2	nd	New Zealand, Lyttelton Harbour	[118]
	S	<0.4–9.9	nd		
	W	5.0–277.9	4.0–1721	Poland, Gulf of Gdańsk (southern Baltic Sea)	[119]
	S	2.64–60.20	0.08–1001		
**Drinking water**		6–53	-	Poland	[108]
		2.6–6.2	1.7–3.9	Serbia, rural area	[120]
		2.5–35.6	1.2–7.9	Serbia, urban area	[120]

nd–not detected; IW-influent; EW–effluent; W–water; S-sediment.

The presence of NP in surface waters is correlated with anthropogenic activity and wastewater discharge from STW, industrial plants, municipal wastewater treatment plants, and rainwater discharges [46,47]. NP concentrations in rivers are subject to seasonal fluctuations with higher concentrations in summer due to increased microbial activity at higher temperatures leading to increased degradation of ethoxylates nonylphenol [121,122]. Other factors such as river flow rate, sedimentation rate, and particle size also influence the rate of degradation. NP occurs in the aquatic environment in quite different concentrations in surface waters, from several dozen ng/L to several dozen mg/L [41].

Groundwater is of particular interest as it accounts for about twenty percent of the world’s freshwater supply and is extremely susceptible to contamination by various pollutants as a result of urban activities. NP concentrations in groundwater were incredibly low [40,123,124]. The microbial decomposition of NP in aquifers is limited due to the low temperature conditions prevailing there: low temperature, low C, and low O_2_ content). The processes controlling the entry of pollutants into groundwater are sorption and biodegradation [46,98].

Ethoxylates nonylphenol (NPEO) ends up in significant amounts in wastewater treatment plants, where they biodegrade into several by-products, including nonylphenol, which is more resistant than the parent compound [46,125]. NP is the major degradation product, does not undergo further transformation, and is highly adsorbed in the sludge; therefore, it is often found in higher concentrations in wastewater than in tributaries [46,125,126]. In most cases, it is the strong sorption of pollutants, and not the microbiological activity, which limits the rate of biodegradation [127]. On the rate of biodegradation NPEO, the main sources of NP in WWTPs, is affected by many factors e.g., temperature, NP isomer present in the environment, oxygen availability, pH and additives of yeast extracts, surfactants, aluminum sulphate, acetate, pyruvate, lactate, manganese dioxide, iron chloride, sodium chloride, hydrogen peroxide, heavy metals, and phthalic acid esters [128,129,130].

Conventional physicochemical wastewater treatment methods have not been shown to be effective in removing endocrine disruptors such as NPs due to their low molecular weight [27,131]. Novel purification techniques, including advanced oxidation methods, UV [132], adsorption (powdered activated carbon (PAC), and granular activated carbon (GAC)), ion exchange and membrane filtration (ceramics, polymers, and zeolites) are still under analysis [133,134].

One of the other methods of NP removal is to use cells and enzymes. Tanghe et al. in 1999 first described the *Sphingomonas* strain that degraded NP [135]. Since then, microorganisms involved in the biodegradation of NPs in the aquatic environment have been discovered (*Sphingobium, Pseudomonas, Pseudoxanthomonas, Thauera, Novosphingonium, Bacillus, Stenotrophomonas, Clostridium, Arthrobacter, Acidurvorax, Rhizobium, Corynebacterium, Traynebacterium, Rhodococcus, Candida, Phanerochaete, Bjerkandera, Mucor, Fusarium and Metarhizium*) [34,130,136,137,138,139]. NP-degrading microorganisms were isolated from municipal sewage treatment plants from sludge, sewage sludge and activated sludge under aerobic and anaerobic conditions. Appropriate pH value, temperature, and the level of aeration of sewage sludge contribute to the increase of microbiological activity, and thus increase the degradation of NP.

Anaerobic degradation of NP has only recently been shown. Half-lives of anaerobic degradation ranged from 23.9 to 69.3 days. The rate of anaerobic degradation of NP was enhanced by increasing the temperature and adding yeast extract or surfactants [41,98,140]. Bacterial strains are effective in improving NP biodegradation and short-chain fatty acid accumulation. The species *Propionibacterium*, *Paludibacter*, *Proteiniphilum*, *Guggenheimella*, *Lactobacillus*, *Anaerovorax* and *Proteiniborus* correlated with the short-chain fatty acids synthesized as a result of NP degradation [137,141].

Some bacteria and fungi produce laccase, a multifunctional enzyme. Laccases are well-known enzymes that have found application in bioremediation, both as free and immobilized enzymes. Thanks to their broad range substrate specificity, laccases are able to remove xenobiotics, including EDCs. Therefore, the white rot fungi (WRF) are a promising tool in the above-mentioned elimination of EDCs during wastewater treatment processes [53]. The advantage of fungi over bacteria in lignin mineralization results from the production and secretion of laccase-non-specific enzyme outside the cell, which gives fungi access to non-polar and insoluble substances, operation over a wide range of temperatures and pH values, and the developing fungal hyphae also make it possible to reach contaminants inaccessible to bacteria [53,142,143]. The laccase from WRF like *Pleurotus eryngii*, *Trametes versicolor* or *Phanerochaete chrysosporium* can adhesion to alkylphenols, because these compounds have functional groups such as amino, hydroxyl, or alkyl groups in its chemical structure, acting as electron donors for oxygenases [53,144,145,146,147,148].

### 3.2. BPA

BPA is a monomer in the production of polycarbonate plastics and epoxy resins, and as an additive in the production of PVC coatings is added to various plastics as a plasticizer. These materials are used in food storage containers, water, baby bottles, food, and drink cans. Under the influence of elevated temperature, BPA may migrate from containers to food and beverages [44,149]. BPA-derived monomers, especially bis-GMA (methacrylate bisphenol–glycidyl) are used in dental materials, from where they can be released [150].

The release of BPA to the aquatic environment takes place in several ways, including from production plants, from wastewater as a result of incomplete treatment, or in physicochemical and biological processes in treatment units, from leachate from landfills, as well as from leaching from discarded BPA-based products [17,19,67,86,105]. The problem is also sludge from recycled paper, the production of which uses BPA as a reactive agent. These sediments are used as fertilizers in agriculture, eventually contaminating the groundwater with BPA [151]. BPA also enters groundwater through its release to landfill leachate [40,152,153,154,155].

The results showed that the leaching of BPA from polyethylene microplastics (LDPE) and polycarbonate (PC) was 2.68 μg/g and 14.45 μg/g, respectively [13]. Studies have shown that diffusion of BPA in the environment is related to the hydrolysis of PC: processing time for polycarbonate bottles in the presence of water is only a few (3–7) years [79].

BPA concentrations in surface water vary depending on the location, sampling period and depth of sampling. Studies in which both water and sediment were collected indicate significantly higher BPA concentrations in the sediments than in the upper water column, which is related to the slowing down of biodegradation processes in anaerobic environments [15,16,152,156]. Rivers in Europe and North America with higher BPA concentrations detected are commonly associated with production facilities [16,79]. Although BPA dissolved in surface water has a short half-life due to photo- and microbial degradation, its metabolites can persist much longer [14]. The BPA metabolites can be toxic for aquatic organisms [67]. Like BPA, its metabolites are xenoestrogens, e.g., 4.40-dihydroxy-methylstilbene and 4-methyl-2,4-bis (4-hydroxyphenyl) pent-1-ene (MBP), whose estrogenic activity levels exceed BPA levels respectively 40- and 300-fold [157,158].

Observed BPA concentrations in oceans and estuaries are low compared to freshwater systems (Table 1). It is related to the fact that sewage from municipal and mixed municipal-industrial wastewater treatment plants is the main source of environmental BPA, and its runoff goes to rivers [15,17,22,159,160,161]. BPA is leached faster in marine systems than in freshwater systems [162,163], and microbial degradation may be slower [24,164]. Moreover, the bioavailable fraction of dissolved BPA may increase with salinity [163].

The poor biodegradability of BPA in nature leads to the contamination of surface and groundwater. We can remove BPA using the following methods: photodegradation (TiO_2_) is the most commonly studied and uses photocatalyst, adsorption (natural adsorbents, carbon and graphene, clay, nanomaterials, and composite materials), biodegradation with the use of microorganisms, or phytoremediation [96,165,166,167,168].

The main source of microorganisms in biodegradation is activated sludge from WWTPs. *Bacillus thuringiensis*, *Pseudomonas putida* YC-AE1, *Sphingomonas paucimobilis* FJ-4, *Lactococcus lactis*, *Bacillus subtilis* and many more bacteria are capable to use BPA as substrate in their metabolism [79,169]. Among fungi, BPA degradation was observed in *Saccharomyces cerevisiae* and WRF [170,171]. WRF converts BPA into a much less reactive substance in enzymatic oxidation. The oxidized form of BPA does not bind to ERα dependent estrogen receptors. This may be due to laccase oxidation of both BPA hydroxyl groups. The product obtained by laccase catalyzed oxidation of BPA is 2,2-bis (4-phenylquinone) propane [172].

## 4. Laccase-Assisted NP and BPA Biodegradation

Laccases are a multi copper metalloenzymes commonly found in plants, fungi, bacteria, and invertebrate animals. These enzymes, first described back in the 19th century, belong to the class of oxidoreductases [EC 1.10.3.2], the polyphenol oxidases [173,174,175]. Together with ferroxidases, ascorbate oxidases, bilirubin oxidases, and lactase-like multicopper oxidases, they are included in the multicopper oxidases (MCOs) family of proteins [176]. These are proteins with diverse functions in living organisms.

The best described and characterized fungal laccases are most often included in the ligninolytic enzyme complex, whose main function is to depolymerize lignin present in plant cell walls. In addition, fungal laccases engage in morphogenesis and sporulation, in phytopathogenic species they take part in the detoxification of toxic substances synthesized by the plant immune system [177,178]. For example, bacterial laccases can help protect the spore from UV radiation and hydrogen peroxide [179]. Among other things, plant laccases participate in the polymerization of lignin [180]. In insects, on the other hand, laccases oxidize pyrocatechins in the cuticle to the corresponding quinones, which catalyze protein cross linking reactions and are involved in the sclerotization of the cuticle [181]. Differences between laccases also exist in the structure of the protein molecule. Fungal and plant laccases are glycoproteins. The glycosylation process affects copper ions, providing the enzyme with thermal stability and protection from proteolytic degradation. Intracellular bacterial enzymes are often not glycosylated [182,183].

### 4.1. Laccase Reaction Mechanism

In the catalyzed reaction, laccases use molecular oxygen as an electron acceptor and produce a water molecule due to oxygen reduction [178,184]. The mechanism of the reactions catalyzed by laccase, is based on the formation of a product as a result of the oxidation of substrate molecules to oxygen radicals with the simultaneous reduction of an oxygen molecule to two water molecules according to the general notation:4RH + O_2_ → 4R- + 2H_2_O(1)

These are enzymes with broad substrate specificity, their natural substrates being phenolic and non-phenolic derivatives, including di- and methoxyphenols, phenolic acids, aromatic amines, and other lignin-related compounds [177]. The chemical structure of substrate is an extremely crucial factor for the reaction to continue with high efficiency. The preferred functional groups are amine, hydroxyl, carboxyl, methoxyl, and sulfonic groups, which make the oxidation of the substrate by the enzyme faster. Compounds with a more complex structure, which do not have functional groups, need the presence of reaction mediators (mediating compounds) in the exchange of electrons between laccase and the complex substrate [185,186]. Laccase forms multimeric complexes, which can be di-, or tetrameric proteins. Each monomer has four copper atoms with distinctive characteristics, interacting with amino acids to form the active center. Type 1 paramagnetic cooper (T1 Cu), a blue copper, transfers electrons during the substrate oxidation reaction. Type 2 (T2 Cu), non-blue copper is also a paramagnetic atom and engages in electron transfer. The two diamagnetic copper atoms that make up type 3 (T3 Cu) are a spin-coupled copper-copper pair (T3α and T3β), responsible for binding the oxygen molecule. A copper atom of type 2 and two copper atoms of type 3 form a three-atom complex, through which the binding and reduction of molecular oxygen to water takes place [187,188]. The involvement of individual copper atoms in the next reaction steps is explained in selected review papers, for example by Ren at al. (2021) [189].

Three types of reactions catalyzed by laccase have been described. The first involves direct oxidation of the substrate and does not require the presence of mediators. These are oxidation reactions of simple organic compounds such as mono-, di- or polyphenols or derivatives having amine, carboxyl, methoxyl, or sulfonic functional groups. Such reactions often occur in the environment, and are characteristic of cell wall regeneration at the site of plant tissue damage. The second type of reactions are also oxidation reactions but require the presence of supporting mediators, which are low-molecular compounds having specific functional groups (e.g., -NO, -NOH). An example of a reaction belonging to this group is the decomposition of lignin by white wood rot fungi. The third and final type of reactions catalyzed by laccases are coupling reactions, involving the synthesis, formed in the oxidation of phenolic substrates, of unstable and reactive radicals. Such a reaction mechanism allows the formation of new phenolic structures in non-enzymatic processes. An example of such a reaction is the transformation of toxic compounds present in industrial wastewater [173,177,190]. A property of laccases that is extremely important because of its application potential is the redox potential (E0) of the type I copper atom. According to this criterion, laccases were divided into three categories: enzymes with low, medium, and high redox potential. Bacterial and plant laccases usually have a low redox potential, in contrast to fungal laccases, which most often fall into the medium and high potential categories. Enzymes with medium redox potential, are produced by Ascomycota and Basidiomycota fungi, while those with high potential are synthesized by white rot fungi [191]. The value of the redox potential (E°) of laccases is directly related to the energy required to remove an electron from a reducing substrate, This is why laccases of such fungi are of particular interest in biotechnology, as they are capable of oxidizing substrates with high E° (E° > 400 mV), such as EDCs [191,192,193].

The mentioned catalytic abilities of laccases have great application potential, especially in the degradation of xenobiotics. Considering the aspect of bioremediation, an attention is paid to laccases of bacterial and fungal origin. Particularly dangerous for human health, regarding water contamination, is a group of compounds known as EDCs. It has been proven that in the decomposition of these compounds, laccases, which are the subject of this work, play an essential role [194,195]. The biochemical properties of selected bacterial and fungal laccases and their participation in the degradation of NP and BPA as representatives of EDCs showed in Table 2.

**Table 2 microorganisms-10-02236-t002:** Bacterial and Fungal Laccases Degrading NP and BPA compounds.

Alkylphenols	Microorganism	Culture Conditions/Catalysis Conditions	Mode of Treatment	Degradation Efficiency (%)	References
**BPA**	*Pleurotus ostreatus* HK 35	28 °C for 48 h	fungal culture in a trickle-bed reactor	>70	[196]
	*Trametes versicolor*	25 °C > 15 h, pH 5	liquid cultures	>90	
					[197]
	*Funalia trogii*	50 °C, 2 h, pH 5.5	crude extract	100	[198]
	bacterial consortium (BCC1) isolated from the sediment of the Rio Grande River	32 ± 2 °C, stationary culture, 72 h	liquid cultures	100	[199]
	*Pseudomonas putida* G320	30 °C, stirring speed 150 rpm, 96 h	liquid cultures	97	[200]
	*Coriolopsis polyzona*	50 °C, pH 5.0, 4 h	crude enzyme	>95	[143]
	*Pycnoporus sanguineus* CS43	25 °C, pH 5.0	filtered culture supernatant	93	[201]
**NP**	mitosporic fungal strain UHH 1-6-18-4 isolated from NP -contaminated river water	14 °C, pH 4.0, stirring speed of 120 rpm	crude enzyme	63	[202]
	*Cerrena unicolor* C-139	30 °C, pH 5.0	immobilized laccase onto silica beads by covalent binding	40	[203]
	*Thielavia* sp. HJ22	35 °C, 150 rpm in the dark, 8 h	liquid cultures	95	[204]

As we can see in Table 2, laccases with potential use in removing alkylphenols such as NP and BPA are mainly produced by fungi. Most of them belong to mesophilic species, developing in the range of optimal temperatures of 25–35 °C, which increases their usefulness in removing environmental pollutants on a large scale. The main problem may be obtaining the appropriate pH value for the effective operation of laccase and fungal growth, since industrial and municipal wastewater has much higher pH values. Some of the microorganisms were able to be completely removed BPA or NP from the environment by producing exogenous laccases under appropriate conditions.

### 4.2. Immobilization–Method for Improving the Properties of the Enzyme

According to the literature, free as well as immobilized enzymes of fungal, plant, and bacterial origin are used in enzymatic degradation of EDCs [189,205,206]. The efficiency of the bioprocesses depends on the selection of appropriate catalytic tools and optimization of reaction conditions. Enzyme proteins are not always sufficiently stable against organic solvents, so methods are still being looked to increase the efficiency and stability of biocatalysts. This is especially important for the efficiency of catalysis in aqueous environments. One such technique to improve the efficiency of catalysis of xenobiotic decomposition is laccase immobilization. Immobilization is a widely used method in biotechnology to help control of reaction conditions. Immobilized enzyme is less prone to autoinactivation and can be used several times, which reduces the cost intensity of such a process. Thanks to immobilization, it is possible to change some properties of enzymes e.g., shifting the optimum pH to values more convenient for the technological process, often increasing thermostability.

The most basic method of immobilization is adsorption onto the surface of solid ion exchangers. The enzyme binds to the carrier through electrostatic interactions, hydrogen bonds, or van der Waals interactions [189,207,208]. Various natural and synthetic carriers have been used for laccases: porous glass beads [209], green coconut fibers [210], and nanofiber membrane [211]. In this way, laccases used in degradation were immobilized. These are mainly fungal enzymes derived from *Trametes versicolor* or *Trametes hirsuta* [189,212,213].

Techniques in which covalent bonds are produced between the enzyme and the carrier, using functional groups such as amine, imidazole, or phenyl groups and others, are also used for enzyme immobilization. An economical yet effective solution for covalent immobilization of laccases seems to be the use of natural, carbon-based carriers such as biochairs. These are carbon products obtained after pyrolysis, with extremely high specific surface area. Such natural, readily available carbon carriers are often further changed with organic acids to increase the availability of functional carboxyl groups. Glutaraldehyde is often used as a binding agent, and pine wood shells from almonds and even pig manure can be used as an organic carrier [214].

An interesting example is the use of laccase immobilized on biochar modified with magnetic nanoparticles to degrade bisphenol A in an aqueous environment. The laccase immobilization process itself was a three-step process involving adsorption, precipitation, and crosslinking with glutaraldehyde. According to the authors’ results, such an immobilization technique ensured high efficiency of *T. versicolor* laccase in the degradation of bisphenol A [215].

Bisphenol A was successfully degraded by laccase trapped in a matrix of polyethylene glycol-based hydrogel microparticles. Hydrogels of diverse types including agar-agar, gelatin, or combination in poly (acrylamide/crotonic acid)/sodium alginate, poly (acrylamide/crotonic acid)/K-carrageenan have been successfully used for years in enzyme immobilization [216].

Emulsion polymerization of a matrix consisting of poly (ethylene glycol) diacrylate (PEGDA), and poly (ethylene glycol) methacrylate (PEGMA) was supported by UW radiation. The literature data show that such immobilization ensured enzyme stabilization and efficient bisphenol A transformation [217]. It is also worth mentioning here the immobilization technique developed in recent years using nanofibers produced by the electrospinning method as carriers. Immobilization of *T. versicolor* laccase by adsorption and encapsulation using poly (l-lactic acid)-co-poly (ε-caprolactone) (PLCL) electrospun nanofibers as carriers supplied efficient degradation of naproxen and diclofenac [218].

## 5. Conclusions

Long-term exposure to even low concentrations of EDCs and the wide range and complexity of their mechanisms of action have an enormous impact on human and animal health. The scope of these impacts depends on many factors, including type of cells and tissues exposed to contact with hazardous chemicals, circadian rhythms, changes in seasons, stage of development, or gender [1].

Many factors influence the bioremediation of EDCs. The organic content of sediments was one of the important determinants of the adsorption process, especially for ethoxylates shorter chain NP [219,220]. NP concentrations in sediments were higher than in surface water [41]. NP remediation is possible by physicochemical and microbiological methods. Industrially produced technical grade NP is less biodegradable due to branched alkyl quaternary carbon in more than 85% of NP isomers [99,221].

BPA is ubiquitous in the environment due to its continuous release. Release may occur during the production, transport, and processing of chemicals. Post-consumer emissions result from the discharge of wastewater from municipal wastewater treatment plants, leaching from landfills, incineration of household waste, and the natural degradation of plastics in the environment [14]. Thus, BPA removal from the natural environment is an increasing worldwide concern. Biodegradation is expected to be the dominant process for removal of BPA from the aquatic environment.

EDCs penetrate aquatic ecosystems through the wastewater treatment system, industrial waste, municipal waste, agriculture, aquaculture, direct release of pharmaceuticals, chemicals, and indirect releases from sources such as rainwater runoff [222,223,224]. As a consequence, EDCs are absorbed, accumulated, and biomagnified in water, sediment, and biota. These harmful chemicals are periodically released into water bodies as they undergo biogeochemical processes [225]. The use of green methods that do not require toxic and hazardous materials should be a special aspect in the removal of NP and BPA.

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
