# Peer review of "Endocrine Disrupting Compounds (Nonylphenol and Bisphenol A)–Sources, Harmfulness and Laccase-Assisted Degradation in the Aquatic Environment"

_microorganisms, 2022, doi:10.3390/microorganisms10112236_

Round 1
Reviewer 1 Report
The paper is the literature review on the sources and degradaion in the aquatic environment of two alkylphenol compounds: nonylphenol and bisphenol A. The authors discuss the properties and chemical reactions of these two compounds, and their influence on the human health. The paper may be published with minor revisions:
1) the title is too wide, it should be specified according to the paper contents;
2) there are some misprints and repeated words in the text, the paper should be proofreaded.
Author Response
Dear Reviewer 1 ,
Thank you very much for the comments you sent. Of course , we make corrections to the text.
Any corrections in the manuscript are marked with the "Track Changes" function.
- The title was changed as suggested by the reviewer. In the current version it reads as follows: Endocrine disrupting compounds (nonylphenol and bisphenol A) – sources, harmfulness and laccase-assisted degradation in the aquatic environment.
- The text was thoroughly corrected, misprints and repetitive words / phrases were removed.

Reviewer 2 Report
Dear authors,
This article mainly describes the source, harm and degradation of alkylphenols. I think the degradation of alkylphenols in water environment is an important topic. This makes the manuscript of great significance in basic research.
Some of my remarks.
1. The abbreviation for endocrine active compounds is not EDCs, it should be EACs.
2. According to the title, the main purpose of this paper is to discuss alkylphenols and their degradation, but the whole paper introduces EACs in a large number of pages, which is messy, although it has been stated that alkylphenols belong to EACs. It is recommended to focus more on alkylphenols,or make changes to the title.
3. As a review, while citing literature, it lacks analysis and combing. Many sentences are missing full stops.The reference format is also not uniform.
4. The BPA structure is not discussed in Section 2.2.
5. WWTPS in Table1 should be clear about what they are.
6. Table2 simply lists some data, lacking analysis results and the ability of laccase to degradation EACs .
7. The introduction of the immobilization method of laccase in Section 4.2 is not detailed enough.
Author Response
Dear Reviewer 2 ,
Thank you very much for your insightful review. All comments have been incorporated into the text .
Any corrections in the manuscript are marked with the "Track Changes" function.
- In a large part of the article and the works cited in it, the authors used the abbreviation EDCs for Endocrine Disrupting Compounds. The nomenclature was standardized throughout the text and in lines 609,615 the wording Endocrine Active Compounds (EACs) was replaced with Endocrine Disrupting Compounds (EDCs). In verse 620, the subject is narrowed down to NP and BPA.
- The title was changed as suggested by the reviewer. In the current version it reads as follows: Endocrine disrupting compounds (nonylphenol and bisphenol A) – sources, harmfulness and laccase-assisted degradation in the aquatic environment.
- The text was thoroughly corrected, misprints were removed. The reference format has been revised.
- In section 2.2. added description of BPA structure in lines 263-265.
- In Table 1, the abbreviation WWTPs has been replaced by the full name, i.e. Wastewater Treatment Plants.
- Table 2 has been re-titled to focus on NP and BPA. Data was supplemented with the efficiency of degradation of these compounds. Misprints and references are fixed. Below the table there are conclusions from the analysis of the data contained in Table 2 (line 621-628).
- The subsection on laccase immobilization has been expanded and enriched with new examples of techniques.

Round 2
Reviewer 2 Report
There are still some spelling problems in the text,such as the”possessoestrogenic " in the abstract. Many sentences are missing full stops...
Author Response
Dear Reviewer 2
Thank you very much for your comments, we have corrected them carefully, in addition we have revised the English language
Best regards
